# The alteration of uterine microbiota participated in the activation of the decidual inflammatory response in early spontaneous abortion

Ping Liu[1,2]☉, Ge Chen[1,2]☉, Shitong Zhao[1,3], Linglingli Kong[1,2], Xin Liao[2,4], Meng Cheng🆔[1,2]*

**1** Department of Gynecology and Obstetrics, West China Second University Hospital of Sichuan University, Chengdu, Sichuan, China, **2** Key Laboratory of Birth Defects and Related Diseases of Women and Children, Sichuan University, Ministry of Education, Chengdu, Sichuan, China, **3** Department of Gynecology and Obstetrics, HanYuan People's Hospital, Yaan, Sichuan, China, **4** Department of Operating Room Nursing, West China Second University Hospital of Sichuan University, Chengdu, Sichuan, China

☉ These authors contributed equally to this work and share first authorship.
* ttt-cheng@163.com

## Abstract

### Background

Early spontaneous abortion (ESA) is one of the most common clinically recognized pregnancy complications. While multiple factors such as embryo abnormalities and maternal conditions may contribute to ESA, early identification and screening of maternal risk factors are increasingly important to explore the potential etiologies and improve prevention and treatment strategies for ESA. This study investigates the changes in uterine microbiota and the decidual immune response in ESA patients without embryo abnormalities.

### Methods

ESA patients without embryo abnormality and artificial abortion (AA) controls were enrolled for clinical characteristics analysis. The decidual endometrium was subsequently collected for histological evaluation and inflammatory indicator detection. Moreover, 16S rRNA gene sequencing of uterine secretions was performed to investigate the differences in uterine microorganisms between the ESA and AA groups.

### Results

Clinical analysis showed higher inflammatory response with elevated neutrophil counts in ESA patients. The increase in leukocytes, including neutrophils, was positively correlated with ESA. ESA patients presented significantly increased IL-1β expression in decidual stromal cells. 16S rRNA gene sequencing revealed greater diversity in the uterine microbiota of the ESA group, which presented decreased Lactobacillus abundance and increased abundance of other bacteria at the genus and species levels.

**Data availability statement:** All relevant data are within the paper and its Supporting Information files.

**Funding:** 'This research was funded by the National Natural Science Foundation of China (Website: https://www.nsfc.gov.cn/). MC received the grant (No.82101717). The funder had no role in study design, data collection and analysis, decision to publish, or preparation of the manuscript.'

**Competing interests:** The authors have declared that no competing interests exist.

## Conclusions

Changes in the uterine microbiome are likely related to inflammatory response and lead to early pregnancy loss.

## Introduction

Spontaneous abortion is one of the most common clinically recognized pregnancy complications in the general population, with an incidence of approximately 15%, with more than 80% of cases considered early spontaneous abortion (ESA) as they occur before 12 weeks [1]. The consequences of miscarriage are both physical and psychological, such as infection, bleeding, depression, anxiety, and posttraumatic stress disorder [2]. Many factors may contribute to ESA, including embryo chromosomal abnormalities and maternal factors, such as immune dysfunctions, endocrine disorders, uterine dysplasia or malformation, antiphospholipid syndrome and thrombophilias [1–4]. With the wide application of preimplantation diagnosis technology, early screening and identification of maternal risk factors has become increasingly important to further investigate the potential aetiologies, prevention, and treatment of spontaneous abortion.

Studies [5–8] on the role of the vaginal microbiota in early pregnancy miscarriage have shown that patients with ESA have reduced *Lactobacillus* spp.-dominated vaginal microbiota and greater diversity and richness of the bacterial community. The endometrial microbiota plays a crucial role in several obstetrical complications, increasing attention has been focused on the uterine microenvironment to explain the causes of miscarriage [9,10]. Altered uterine microbiome composition reportedly contributes to poor reproductive outcomes, such as failed embryo implantation [11], early spontaneous abortion [12] and recurrent spontaneous abortion [13]. However, clinical research about the upper genital tract with ESA is limited, for the difficulty in collecting samples and the possibility of contamination.

Alterations in the composition of uterine microorganisms can induce an inflammatory response, which is closely linked to the adhesion of the blastocyst to the epithelial endometrial wall. Researchers have shown that successful pregnancy requires the involvement of an inflammatory response, with alterations in cytokines and chemokines [14]. However, over-activated immune response may be responsible for spontaneous abortion. Endometrial expression of the pro-inflammatory cytokines (IL-1β, TNF-α, IFN-γ and TGF-β1) and prostaglandin E2 (PGE2) are upregulated and the expression of anti-inflammatory cytokines (IL-4 and IL-10) and angiogenesis-associated cytokines (IL-2, IL-6, and IL-8) are downregulated in idiopathic recurrent spontaneous miscarriage women compared with normal fertile women [15].

Are changes in decidual inflammatory cytokines associated with alterations in the uterine microbiota? Do the changes in inflammatory cytokines and the uterine microbiota contribute to early spontaneous abortion? In this study, we analysed the clinical characteristics of ESA without embryo abnormalities compared with those of normal pregnancies to determine the effects of the composition of uterine microbial communities and inflammation-related indicators and explore the high-risk factors of ESA.

## Materials and methods

### Study design

This single-centre clinical observational study was conducted at West China Second University Hospital of Sichuan University. Ethical approval for the study was granted by the Ethics Committee of West China Second University Hospital of Sichuan University (No. 2023-350).

The patients/participants provided their written informed consent to participate in this study. The study consisted of two parts. The first part was a retrospective analysis, in which the clinical characteristics and laboratory test results of ESA (early spontaneous abortion) and AA (artificial abortion) patients were collected, compared, and analysed. The requirement for informed consent was waived by the ethics committee for the retrospective analysis of the medical records, and the data were anonymised to remove any personally identifying information. The second part was an observational study, where decidual tissues and uterine secretions from ESA and AA patients were collected for histological analysis, cytokine detection, and microbial analysis. Written informed consent was obtained from all patients who provided samples.

## Data collection

The retrospective analysis was started from 26/04/2023. Patients who met the following inclusion/exclusion criteria and underwent outpatient surgical termination of pregnancy between 01/01/2023 and 31/03/2023 were recruited. The included patients were divided into two groups according to the survival status of the embryos at the time of termination. Patients with a nonviable intrauterine pregnancy, such as an empty gestational sac or a gestational sac containing an embryo or foetus with no foetal heart activity, were included in the ESA group. The AA group included patients who chose to terminate unwanted pregnancies with normally developing embryos. Clinical information such as the patient's age, maternal history, gestational sac size, medication use during pregnancy, routine blood, routine leucorrhoea and ultrasound findings were collected.

The inclusion criteria were as follows: 1) Patients fully understood the potential risks and benefits of this research and agreed to participate. 2) Surgical termination of pregnancy up to 10 weeks of gestation, including artificial abortion and early spontaneous abortion. 3) Maternal age between 20 and 35 years. 4) The number of pregnancies was less than or equal to 3 (including the present time). 5) No COVID-19 virus infection during pregnancy.

The exclusion criteria were as follows: 1) early spontaneous abortion without embryo chromosome examination or chromosomal abnormality in the embryo. 2) Patients with abnormalities in the immune system or the use of immunosuppressive drugs, such as hydroxychloroquine, prednisone, tacrolimus, etc. 3) Patients with acute genital tract inflammation, such as all kinds of vaginosis or HPV infection. 4) Two or more consecutive spontaneous abortions. 5) Any pathological disease affecting the endometrial cavity, such as endometrial polyps, submucosal myomas, intramural myomas and adenomyosis.

## Sample collection and tissue preparation

The sample collection of the observational study was conducted between 01/06/2023 and 31/07/2023, and patients who met the inclusion and exclusion criteria were recruited during this period. Prior to the acquisition of the embryo chromosomal results, all samples were properly stored. Sample collection was conducted in a sterile operating room during surgical abortion procedures. To prevent contamination from the vagina and cervix, the operator proceeded to sterilise the vagina and cervix, wipe away any cervical secretions, and dilate the cervical canal in accordance with the surgical procedure. Prior to the collection of the microbiota sample, the cervical canal was disinfected with iodophor and the residual iodophor was removed with a sterile dry cotton swab. Two sterile dry cotton swabs were separately inserted into the uterine cavity to collect uterine secretions for microbiota analysis. These swabs were placed in sterilized tubes and stored at −80°C until DNA extraction. After collecting the microbial samples, the surgical procedure was completed, and decidual endometrial tissues

were harvested and stored. The decidual endometria were stored at −80°C for cytokine analysis or fixed in 4% paraformaldehyde, embedded in paraffin and sectioned for the subsequent haematoxylin and eosin (H&E) staining, immunohistochemical (IHC) staining, and immunofluorescence staining.

## Immunohistochemical staining

IHC staining was conducted using antibodies against CD138 (Zhongshan Jinqiao Biotechnology Co., ZA-0584), MUM1 (Chengdu Zen-Bioscience Co., Ltd., R50091), and IL-1β (Proteintech, 16806-1-AP). Four nonoverlapping areas per section were photographed at 400x magnification. Image analysis was performed using Fuji ImageJ software to calculate the integrated optical density (IOD) and the area of positively stained cells. The mean density of IL-1β-positive protein expression was computed for quantitative analysis.

## Inflammatory status assessment

To evaluate the inflammatory status, we counted the number of CD138- and MUM1-positive cells in four nonoverlapping high-magnification fields per section. Based on the number of CD138-positive or MUM1-positive cells in each field, sections were categorized into grades 0–2, following a previously established protocol [16]. Specifically, Grade 0 was assigned to sections with no positive cells, Grade 1 for sections with 1–4 positive cells, and Grade 2 for sections with 5 or more positive cells.

## Cytokine analysis

Decidual endometria were collected to measure the levels of inflammatory cytokines using a G-Series Human Inflammation Array (Raybiotech), which profiles 40 cytokines. Following the manufacturer's protocol, 100 μl of each sample was added to the wells of the cytokine array and incubated at room temperature for 1–2 hours. After incubation, the samples were washed thoroughly and subjected to subsequent incubation steps as per the provided instructions.

Fluorescence detection was performed using a laser scanner equipped with a Cy3 wavelength (Axon GenePix). The resulting data were analyzed using microarray analysis software (GenePix) and RayBio analysis tools to quantify the levels of the inflammatory cytokines.

## Quantitative PCR assay for IL-1β

Decidual RNA was extracted from samples via the QIAamp RNA Kit (Qiagen) according to the manufacturer's instructions. The sequences of the primers used for IL-1β were as follows: forward, 5'-TGGCTTATTACAGTGGCAATGAGG-3'; reverse, 3'-AGTGGTGGTCGGAGATTCGTAG-5'. The amplifications were performed on a Droplet Digital PCR system (Bio-Rad) with the following 50°C for 15 minutes, 95°C for 3 minutes, followed by 45 cycles of 95°C for 15 seconds and 60°C for 30 seconds.

## Double-immunofluorescence staining

Double-immunofluorescence staining was performed to characterize the IL-1β-expressing cells in the decidua, and the decidual stromal cells were identified with prolactin. The decidual endometrium slices were incubated with primary antibodies overnight at 4°C and then with fluorescent secondary antibodies for 1 h at room temperature. The primary and secondary antibodies used for IL-1β immunofluorescence staining were an anti-IL-1β rabbit polyclonal antibody (Proteintech, 16806-1-AP) and a fluorescein isothiocyanate (FITC)-labelled antibody (Chengdu Zen-Bioscience Co., Ltd., 550037), respectively. For prolactin immunofluorescence staining, a prolactin mouse

monoclonal antibody (Chengdu Zen-Bioscience Co., Ltd., 222677) and a Cy5-labelled secondary antibody was used (Chengdu Zen-Bioscience Co., Ltd., 550047). The slides were embedded in DAPI containing mounting buffer (Vector Laboratories) and then observed under a Leica DM6B fluorescence microscope. IL-1β appeared green, whereas prolactin appeared red.

### 16S rRNA gene sequencing

Total DNA was extracted from the samples using the PowerSoil® DNA Isolation Kit (Qiagen) following the manufacturer's instructions. For the amplification of the full-length 16S rRNA gene, specific primers were designed based on the conserved regions 27F and 1492R. The forward primer (27F) sequence was AGRGTTTGATYNTGGCTCAG, and the reverse primer (1492R) sequence was TASGGHTACCTTGTTASGACTT. PCR amplification of the target region was performed in a 10 μL reaction system using the Solexa PCR kit.

### Bioinformatic analysis

The quality of the resulting sequencing library was inspected to ensure high-quality data. Following quality control, barcode recognition was performed on the high-quality circular consensus sequencing (CCS) sequences. The optimized CCS sequences were clustered at a 97% similarity level using USEARCH (version 10.0) to generate operational taxonomic units (OTUs). Species classification was based on the sequence composition of these OTUs [16,17].

Species annotation and taxonomic analysis were conducted using the Silva database and the RDP classifier. These tools were employed to analyze the diversity of the gut microbiota within the samples.

Alpha diversity metrics, including the Chao1 index and Shannon index, were calculated to assess the richness and evenness of species within each sample. Beta diversity was evaluated using the Bray-Curtis dissimilarity metric to compare the differences in microbial community composition and structure between samples.

Beta diversity analysis was utilized to compare the community composition and structure across different samples. Metastats analysis was performed to identify significant differences in the relative abundance of microbial genera between the ESA and AA groups.

The functional potential of the microbial communities was predicted using PICRUSt2, with functional annotation based on the KEGG database (https://www.kegg.jp/).

### Statistical analysis

Statistical analyses and visualizations were performed using GraphPad Prism version 9.0 (GraphPad Software), SPSS version 27.0, and R version 4.2. Continuous data were compared using Student's t-test, while categorical data were analysed using the chi-square test, with Fisher's exact test applied when necessary. Data are presented as means with standard deviations (SD) or as counts with percentages. Spearman's correlation analysis was conducted to assess relationships between multiple factors. A logistic regression model was used to assess the association between multiple factors. Coefficients (β) were estimated using the maximum likelihood method, with standard errors (SE), z-values, p-values, and 95% confidence intervals (CI) computed to evaluate the significance and effect size of each predictor. A p-value of less than 0.05 was considered statistically significant.

## Results

### ESA patients had a greater inflammatory response than did AA patients

A total of 154 ESA patients and 128 AA patients were initially screened; however, after applying exclusion criteria, 28 ESA patients and 111 AA patients were ultimately included

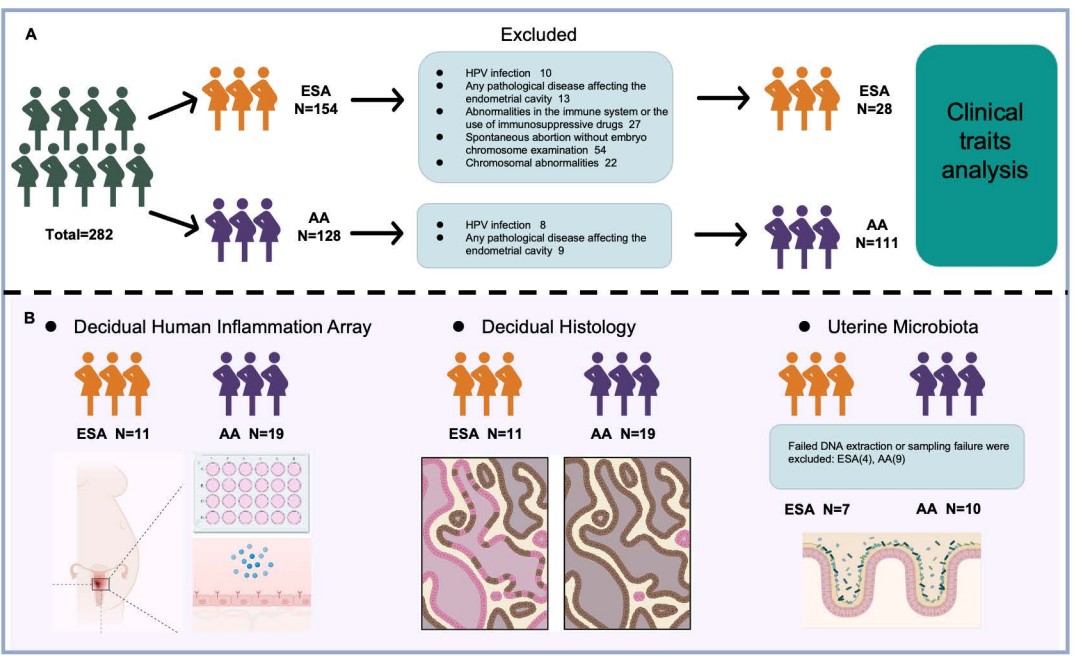

**Fig 1. The flowchart of patient cohorts in study.** *ESA, early spontaneous abortion; AA, artificial abortion.*

in the retrospective clinical trait analysis (Fig 1). Compared with AA patients, ESA patients generally presented a greater inflammatory response, with an increased number of leukocytes, higher counts and percentages of neutrophils and a decreased percentage of lymphocytes (Fig 2A, S1 Fig and Table 1). No obvious differences in age, foetal size, the number or percentage of monocytes, number of lymphocytes, blood platelet count, clarity of leucorrhoea, leukocytes in leucorrhoea, or the level of *Lactobacillus* in leucorrhoea were detected between the two groups (S1 Fig and Table 1). The correlation analysis also indicated that the occurrence of ESA showed positive relation with the number of leukocytes and neutrophils, and negative relation with the number and percentage of lymphocytes (Fig 2B). A logistic regression model was used to assess the association between age, leucorrhea cleanliness, leucorrhea cell content and leucorrhea lactobacillus content (independent variables). Age ($\beta = -0.235$, $p = 0.017$) and leucorrhea cleanliness ($\beta = -1.429$, $p = 0.006$) were significantly negatively associated with the outcome, indicating that increases in these variables reduced the odds of the condition. Conversely, leucorrhea cell content ($\beta = 2.559$, $p = 0.005$) showed a significant positive association, suggesting that higher levels increased the odds of the condition. Leucorrhea bacterial content ($\beta = 0.077$, $p = 0.802$) was not significantly associated with the outcome (S1 Table in S1 Text).

Since the ESA group was sampled with no information on whether embryo chromosome abnormalities were present or not. Therefore, when the embryonic chromosome results were available, some samples in the ESA group were excluded because of embryonic chromosome abnormalities. To further explore the inflammatory changes of ESA patients, decidual tissues of 11 ESA and 19 AA patients were collected. We applied a cytokine array comprising 40 human inflammatory cytokines to detect alterations in the inflammatory status of the decidua of ESA patients. Correlation analysis was performed to investigate the potential association

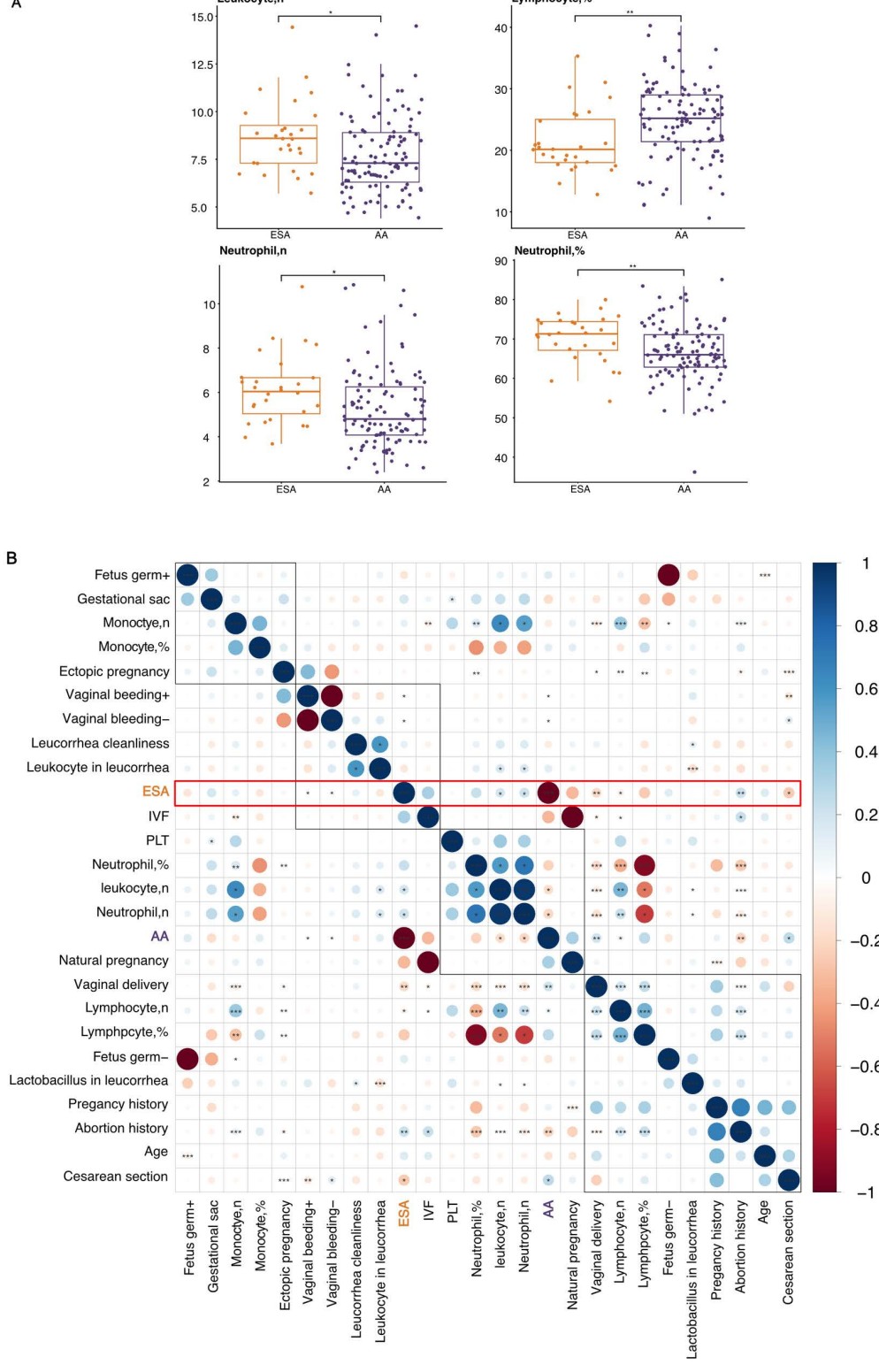

**Fig 2. Clinical characteristics analysis of patient cohorts.** (A) Comparison of the number of leukocytes, percentage of lymphocytes, the number and percentage of neutrophils between ESA and AA groups. Each dot represents an individual ESA (red), AA (blue). (B) Spearman correlation and hierarchical clustering of indicated features for ESA and AA groups. Significance determined by spearman correlation test and student's t test: *p < 0.05, **p < 0.01 and ***p < 0.001. *ESA, early spontaneous abortion; AA, artificial abortion.*

**Table 1. Increased expression of IL-1β was detected in the decidua stromal cells of ESA patients.**

| | ESA (n = 28) | AA (n = 111) | *p* |
|---|---|---|---|
| **Age (Mean ± SD)** | 30.25 ± 2.66 | 28.09 ± 3.91 | 0.086 |
| **Gestational sac (Mean ± SD)** | 2.87 ± 1.28 | 2.31 ± 1.29 | **<0.05** |
| **Fetus size (Mean ± SD)** | 1.25 ± 0.89 | 0.88 ± 0.76 | 0.113 |
| **Leukocyte, n (Mean ± SD)** | 8.68 ± 1.88 | 7.77 ± 2.04 | **<0.05** |
| **Neutrophil, n (Mean ± SD)** | 6.13 ± 1.57 | 5.25 ± 1.79 | **<0.05** |
| **Neutrophil, % (Mean ± SD)** | 70.1 ± 5.98 | 66.5 ± 7.49 | **<0.05** |
| **Lymphocyte, n (Mean ± SD)** | 1.83 ± 0.5 | 1.87 ± 0.48 | 0.0693 |
| **Lymphocyte, % (Mean ± SD)** | 21.38 ± 5.25 | 24.8 ± 6.21 | **<0.05** |
| **Monocyte, n (Mean ± SD)** | 0.87 ± 1.76 | 0.51 ± 0.14 | 0.289 |
| **Monocyte, % (Mean ± SD)** | 6.6 ± 1.04 | 6.75 ± 1.61 | 0.204 |
| **PLT (Mean ± SD)** | 235.39 ± 53.53 | 227.95 ± 52.28 | 0.504 |
| **Leucorrhea test** | **n = 25** | **n = 107** | |
| Leukocyte in leucorrhea | | | 0.508 |
| 0–5 (%) | 1 (4) | 5 (4) | |
| 5–15 (%) | 21 (84) | 78 (73) | |
| 15–30 (%) | 3 (12) | 21 (20) | |
| >30 (%) | 0 (0) | 3 (3) | |
| Leucorrhea cleanliness | | | 0.121 |
| I (%) | 1 (4) | 24 (22) | |
| II (%) | 10 (40) | 36 (33) | |
| III (%) | 9 (36) | 29 (27) | |
| IV (%) | 5 (20) | 19 (18) | |
| Lactobacillus in leucorrhea | | | 0.742 |
| <1 (%) | 1 (4) | 2 (2) | |
| 1–5 (%) | 3 (12) | 20 (18) | |
| 5–10 (%) | 1 (4) | 1 (1) | |
| 10–30 (%) | 12 (48) | 53 (50) | |
| >30 (%) | 8 (32) | 31 (29) | |

between the clinical index and cytokine expression. According to our results, ESA was associated with increased expression of multiple cytokines, including IL-17, TNF-β, IL-1α, IL-1β, IL-7, IFN-γ, IL-2, IL-4, IL-11, and IL-16 (Fig 3). The clustering heatmap shows representative cytokine expression in the two groups. Compared with those in the AA group, I-309, IL-1α, IL-1β, IL-12p40 and MIG were relatively upregulated, whereas MIP-1δ, BLC and TNF RII were downregulated in the ESA group (Fig 4A). Further statistical analysis revealed significantly increased expression of IL-1β in the ESA group (Fig 4B), which was confirmed by quantitative PCR (S2A Fig).

A representative image of IL-1β staining in the decidua is shown in Fig 4C, and the mean fluorescence intensity (MFI) of IL-1β was calculated and compared between the two groups. Consistent with the results of the cytokine test and RNA quantification, the expression of IL-1β was significantly greater in the ESA group (P < 0.05, Fig 4D). IL-1β-positive cells were identified as decidual stromal cells by double immunofluorescence staining for IL-1β and prolactin (Fig 4E). To confirm whether the increased expression of decidual IL-1β was associated with chronic deciduitis, we selected CD138 and MUM1 for histological evaluation. However, no significant difference in CD138- or MUM1-positive staining was found between these two groups (S2D-E Fig).

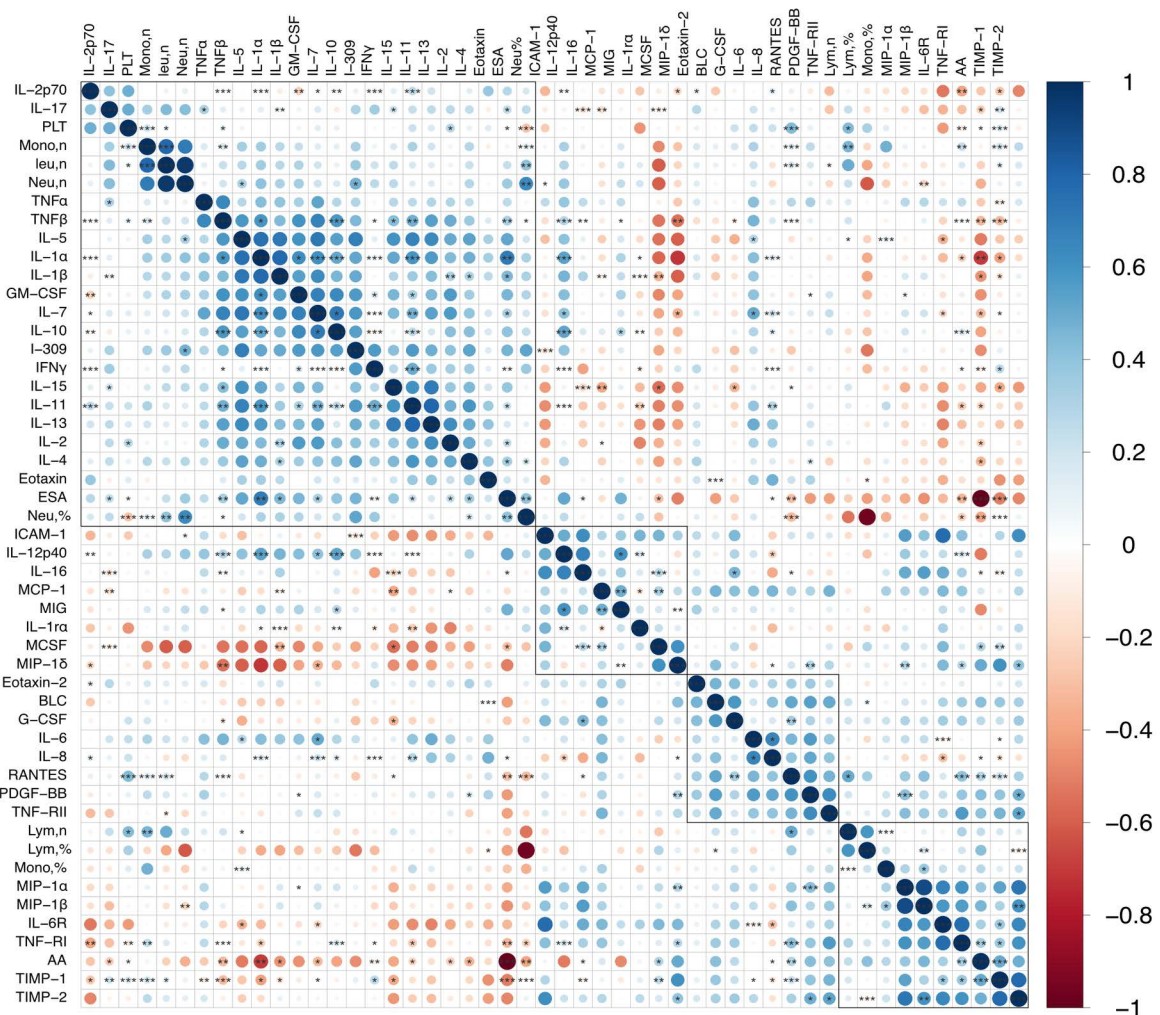

**Fig 3. Spearman correlation and hierarchical clustering of clinical features and 40 inflammation cytokine expression in decidua for ESA and AA groups.** Significance was assessed using Spearmans correlation test and Bayesian analysis. For Bayesian tests, the Benjamini-Hochberg correction was applied to control the false discovery rate across multiple comparisons. Corrected results indicate the strength of evidence after adjusting for multiple hypotheses: *p < 0.05, **p < 0.01 and ***p < 0.001. *ESA, early spontaneous abortion; AA, artificial abortion.*

## ESA patients presented increased diversity of uterine microorganisms

To further investigate whether changes in the uterine microbiome were related with inflammatory response, 16S rRNA gene sequencing was performed. There were 11 uterine swab samples in the ESA group and 19 uterine swab samples in the AA group. Four uterine swab samples from the ESA group and nine uterine swab samples from the AA group were excluded because of DNA extraction failure. A total of 118,522 CCS sequences were produced from the 17 samples (7 ESA and 10 AA). Each sample generated at least 4,224 CCS sequences, and an average of 6,972 CCS sequences were obtained. After clustering, 1,512 and 572 OTUs were obtained from the ESA and AA groups, respectively (S2 Table in S1 Text), and 413 OTUs overlapped between the two groups (Fig 5A). Six OTUs were shared among all the samples in the ESA group, whereas no common OTUs were found across the samples in the AA group (S3 Table in S1 Text, S3A-B Fig).

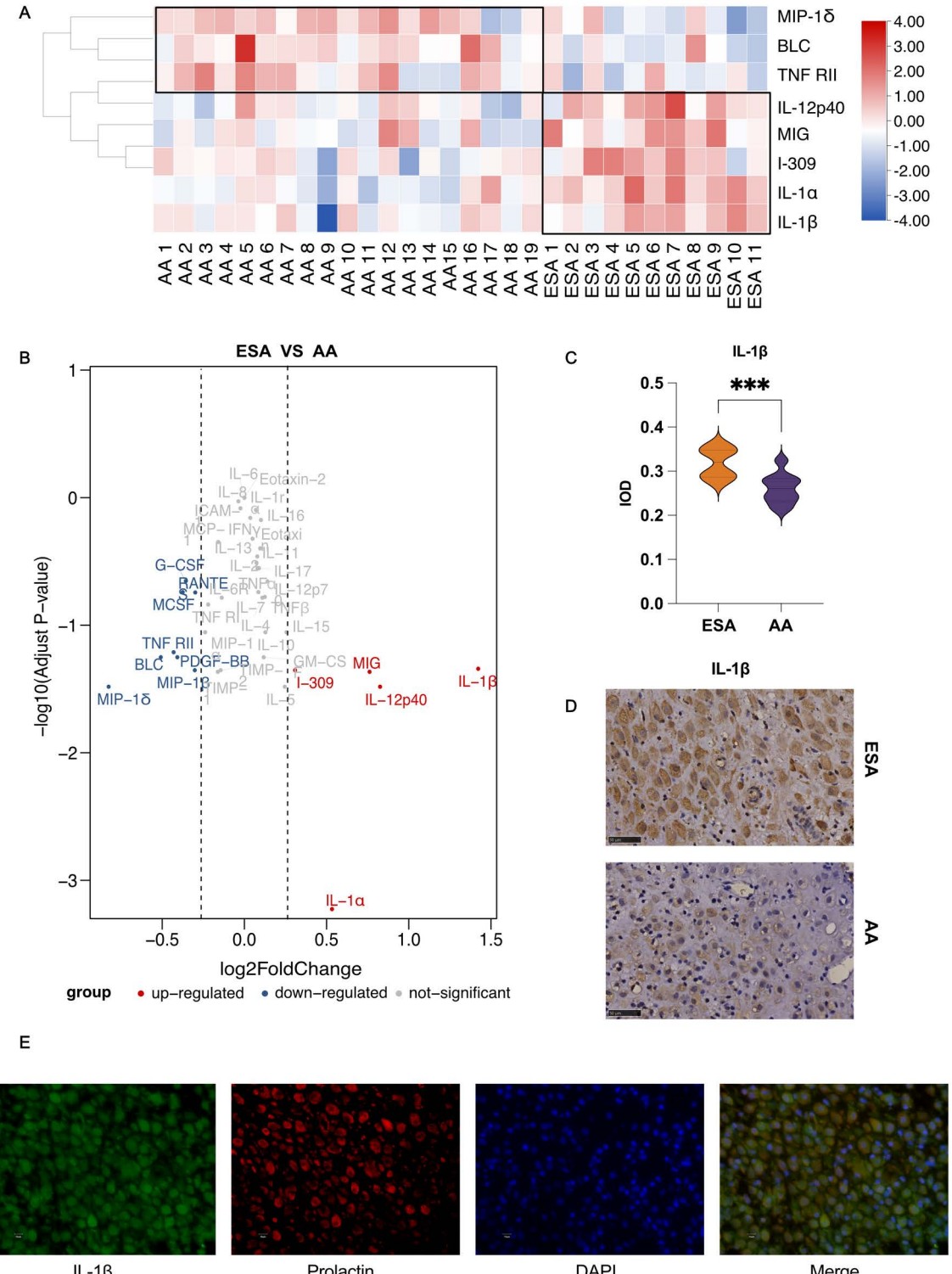

**Fig 4. Analysis of inflammation cytokines expression of decidua between the ESA and AA groups.** (A) Heatmap of the representative cytokines expression in decidua between ESA and AA groups. (B) Volcano plot showed the significantly upregulated or downregulated

cytokines in decidua between ESA and AA groups. (C) The immune-histochemistry staining of IL-1β in decidua of ESA and AA group. (D) The comparison of IL-1β integrate optical intensity (IOD) between ESA and AA group. (E) The double-immunofluorescence staining of IL-1β (green), prolactin (red) and DAPI (blue).

A simple overview of the overall sample distribution was obtained via a clustering heatmap and principal component analysis (PCA) (Fig 5B and S3C Fig). In brief, samples were roughly clustered by group, and samples within groups presented slight differences from each other. The Chao1 index and Shannon index were calculated to estimate the alpha diversity. The Chao1 index (474.69 ± 111.96 for the ESA group and 113.02 ± 26.85 for the AA group) and Shannon index (5.57 ± 0.83 for the ESA group and 1.07 ± 0.54 for the AA group) were significantly greater in the ESA group than in the AA group (Fig 5C-D). ANOSIM, which estimates beta diversity, revealed that the differences between groups were greater than the differences within groups (R = 0.56, P = 0.001) (S3D Fig).

### ESA patients had altered bacterial composition

To investigate whether specific uterine bacteria contribute to the increased risk of ESA, we performed a metastats analysis of the relative abundance between the two groups at the genus level (S4 Table in S1 Text). Among the top ten genera, the ESA group presented a significantly greater abundance of nine genera, including *Acidibacter,* uncultured rumen bacteria, *Lachnoclostridium, Alistipes, Desulfovibrio, Rhizobacter,* unclassified *Longimicrobiaceae, Fimbriimonas* and unclassified *Gaiellales* (Fig 5E). However, *Lactobacillus* was predominant in the AA group. The relative abundance of each individual sample is also shown in S2E Fig. Furthermore, linear discriminant analysis effect size (LEfSe) analyses with linear discriminant analysis (LDA) at the species level were applied to both groups. The bar plot shows the top five most enriched bacterial species in each group (Fig 5F). *Bacteroidota, Bacteroidia, Bacteroidales, Burkholderiales* and *Acidobacteriota* were significantly enriched in the ESA group. The *Lactobacillaceae, Lactobacillus, Lactobacillales, Bacilli* and *Firmicutes* abundances in the AA group were significantly greater than those in the ESA group (Fig 5F).

Functional prediction analysis was conducted on both groups based on Clusters of Orthologous Groups of proteins (COG) functional analysis. COG analysis revealed that the predictive functions of the ESA group were mainly related to amino acid transport metabolism, cell wall/membrane/envelope biogenesis, modification, energy generation and conversion, whereas those of the AA group were related to carbohydrate transport metabolism, replication recombination and repair, translation, ribosomal structure and biogenesis (S3F Fig).

### Discussion

For the past few years, the incidence of early spontaneous abortion has been increasing, and the high incidence of ESA has severely affected many fertile women, families, and society [18]. In this study, we demonstrated that patients with ESA had a greater inflammatory response than did patients with AA. An increased expression level of IL-1β, increased uterine microorganism diversity and altered bacterial composition were found in ESA patients. Alterations in the uterine microbiota and dysbacteriosis may be responsible for these changes and ultimately lead to early spontaneous abortion.

Abnormally activated immune responses may be associated with adverse pregnancy outcomes. Our study found that the markers associated with inflammatory response were elevated in the peripheral blood. In the present study, the cytokine levels in the uterine decidual endometrium were measured to detect alterations in the inflammatory status, and significant

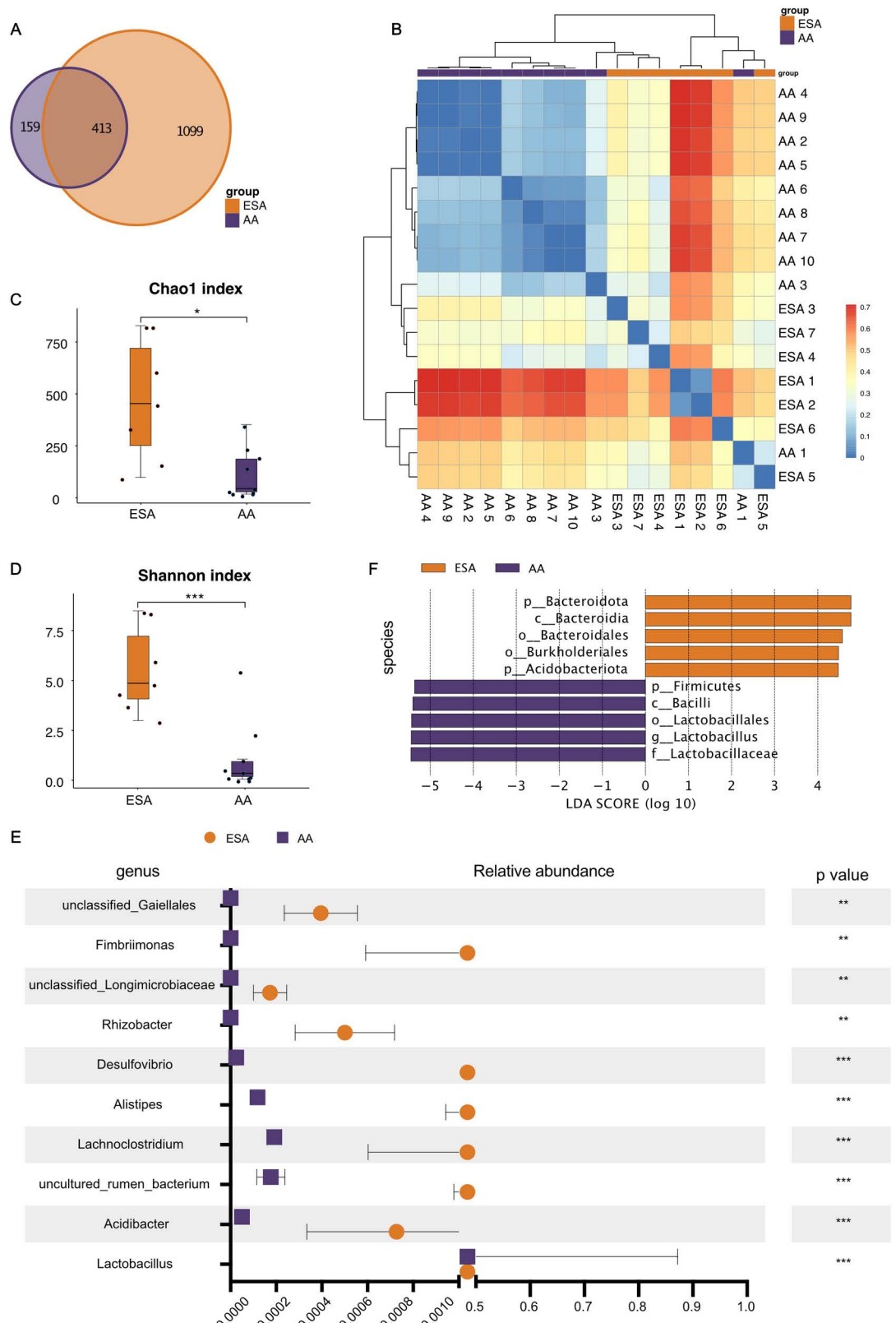

**Fig 5. Analysis of the diversity and relative abundance of the uterine microbial composition in the ESA and AA groups.** (A) Venn diagram between ESA patient and AA groups at OTUs level. (B) Hierarchical clustering heatmap of samples by weighted-unifrac algorithm.

(C) The comparison of alpha diversity between ESA patients and AA groups showed by Chao1 index. (D) The comparison of alpha diversity between ESA patients and AA patients showed by Shannon index. (E) The top10 relative abundance of different bacteria in the two groups at genus level. The middle is the difference in the abundance of bacterial groups within the 95% confidence interval; the right side is the P value, P < 0.05 means the difference is statistically significant. (F) The top5 linear discriminant analysis (LDA) scores obtained from the linear discriminant analysis effect size (LEfSe) analysis of the uterine microbiota in the two groups. Significance determined by Student t test or Wilcox rank sum test: *p < 0.05, **p < 0.01 and ***p < 0.001. *ESA, early spontaneous abortion; AA, artificial abortion.*

upregulation of IL-1β was detected in the ESA group compared with the AA group. IL-1β is a proinflammatory Th1-type cytokine produced by monocytes, macrophages and epithelial cells. IL-1β promotes the proliferation and maturation of B cells, the stimulation of T cells and the activation of natural killer (NK) cells [19]. IL-1β plays an important role in reproductive physiology and has been associated with ovulation, fertilization, decidualization and embryo implantation [20]. A previous study revealed that IL-1β was significantly upregulated in the decidua of spontaneous and recurrent miscarriage placentas [21]. The reason for the elevated level of IL-1β remained to be explained.

The previous study showed that in the patients with endometritis, the cyclic GMP–AMP synthase(cGAS)-STING pathway was activated and the expression of cytokine-encoding genes, including IL-1β, IL-8, IL-6 and IFN-β1 were increased [22]. The level of TNF-α and IL-1β in the uterine cavity with the cesarean scar syndrome group were significantly higher than non-cesarean scar syndrome group. There was an obvious increase in the incidence of chronic endometritis between the cesarean scar syndrome and non-cesarean scar syndrome groups [23]. Some researchers have shown that chronic endometritis is associated with the occurrence of miscarriage, recurrent implantation failure and recurrent spontaneous abortion [24–27]. In our study, decidual endometrial samples were stained with haematoxylin and eosin, and CD138 and MUM1 were detected to identify chronic deciduitis. No difference in the number of plasma cells was identified in the ESA and AA patients which implied involvement of chronic deciduitis in our cases. Goto [26] also reported that no obvious difference in the prevalence of chronic deciduitis between patients with euploid miscarriages and controls. Chronic deciduitis may not be a direct pathogenesis of ESA, but more studies are needed to find the cause for the elevated inflammatory response.

Recent evidence implicates that the microbiota of reproductive tract plays as a key modulator of local inflammatory and immune pathways throughout pregnancy [28]. The euploid miscarriage was associated with a higher prevalence of vaginal *Lactobacillus* spp. depletion compared with aneuploid miscarriage. Higher cytokine levels for TNF-α and IL-1β were more frequently observed in the *Lactobacillus* spp. depleted group [8]. The levels of proinflammatory cytokines (IL-1β, IL-6 and TNF-α) were elevated in women with *Lactobacillus* spp. depleted in the vaginal microbiota group, which could be associated with adverse pregnancy outcomes [29,30]. Microbiome analysis was further conducted to identify microenvironmental change of decidua in ESA patients. Our data revealed greater diversity in the uterine microbiota of the ESA patients, with increased abundances of other bacteria and decreased *Lactobacillus* abundances. These results were consistent with the results of previous studies, 16S rRNA sequencing of the uterine microbiota of women with early spontaneous miscarriage revealed a non-*Lactobacillus*-dominant profile of 76% *Proteobacteria,* 19% *Firmicutes* and 5% *Actinobacteria.* The uterine microbiome in the same woman who subsequently experienced spontaneous miscarriage with euploid embryos had a significantly different profile than that of an early successful pregnancy [6,31].

Clinical studies on the role of the microbiome of the female reproductive tract, especially the upper genital tract with ESA is limited. Existing studies also vary widely depending on the difficulty in collecting samples and the possibility of contamination from vaginal or cervical bacteria.

Our study is a clinical cohort included more than 100 patients with normal pregnancies or early spontaneous abortions. To further validate the elevated inflammatory markers suggested by the clinical trait analysis, we collected samples of decidua from 11 ESA and 19 AA groups and conducted comprehensive analyses. The present study is limited to the correlation analysis between microbiome, inflammatory factors and ESA. Although this study attempted to identify the mechanisms of ESA, it failed to elaborate further due to the complexity of the immune changes caused by pregnancy and abortion, and the limited means to study the microbiotic regulatory mechanisms. However, this is also a common problem in similar studies at present.

## Conclusions

In this research, we performed a detailed study on the uterine microbiomes of patients with ESA and women with AA. Our results highlight the complex interplay of IL-1β at the feto-maternal interface and its potential role in ESA. Changes in the diversity and composition of the uterine microbiota may be correlated with the inflammatory response and imbalance of the immunological response in ESA rather than chronic deciduitis. The study of the uterine microbiome provides insight into the pathogenesis of ESA. The mechanic interactions between microbiome, inflammatory factors and ESA remains unclear. Therefore, a larger sample of the population and investigations of the underlying causal mechanism of ESA are needed in further research to confirm our conclusions. The more investigation of the mechanisms need to be studied in future animal experiments.

## Supporting information

**S1 Fig. Clinical characteristics of patient cohorts.** Comparison of the number of mean age, gestational sac, fetus size, number of lymphocytes, number of platelets, the number and percentage of monocyte between ESA and AA groups. Each dot represents an individual ESA (red), AA (blue). Significance determined by Student's t test: *p < 0.05. *ESA, early spontaneous abortion; AA, artificial abortion.*
(TIF)

**S2 Fig. Histology evaluation of chronic deciduitis.** (A) Comparison of IL-1β RNA expression between ESA and AA by qPCR. (B) Immunohistochemical (IHC) staining and Grading of CD138 in decidua. (C) Distribution of different grade of CD138 staining between ESA and AA. (D) Distribution of different grade of MUM1 staining between ESA and AA. (E) Immunohistochemical (IHC) staining and Grading of MUM1 in decidua. Significance determined by $chi^2$ test and Student's t test: *p < 0.05. *ESA, early spontaneous abortion; AA, artificial abortion.*
(TIF)

**S3 Fig. Analysis of the diversity and relative abundance of the uterine microbial composition in ESA and AA groups.** (A) and (B) OTUs analysis of each sample and group by petal diagram, the central circle represented the number of OTUs common in the same group, and the non-overlapping part was the number of OTUs unique to each sample. (C) PCoA plot of weighted-unifrac distance between ESA and AA groups. (D) Beta diversity by Anosim analysis of weighted-unifrac distance within and between ESA and AA groups. (E) Relative abundances of the uterine microbiota of each sample at the genus level. (F) Clusters of Orthologous Groups of proteins (COG) functional comparison of significantly differentially abundant bacterial in ESA and AA group. *ESA, early spontaneous abortion; AA, artificial abortion.*
(TIF)

**S1 Text. S1 Table Logistic Regression Analysis of Factors Associated with spontaneous abortion. S2 Table Number of OTUs Obtained from Clustering Reads of each sample. S3**

Table Number and taxonomy of shared OTUs of each sample in ESA group. S4 Table Metastats analysis of the relative abundance between the two groups at the genus level.
(XLSX)

## Acknowledgments

We thank Fengqiong Lv, Chang Liu and Qiuyi Wang (all from West China Second University Hospital of Sichuan University, Chengdu, Sichuan) for sample collection and clinical data.

## Author contributions

**Conceptualization:** Ping Liu, Ge Chen, Meng Cheng.

**Data curation:** Ping Liu, Ge Chen, Shitong Zhao, Linglingli Kong, Xin Liao, Meng Cheng.

**Formal analysis:** Ge Chen, Xin Liao.

**Funding acquisition:** Meng Cheng.

**Methodology:** Ping Liu, Shitong Zhao, Linglingli Kong, Xin Liao.

**Software:** Ge Chen, Meng Cheng.

**Supervision:** Meng Cheng.

**Writing – original draft:** Ping Liu, Ge Chen, Meng Cheng.

**Writing – review & editing:** Ge Chen, Meng Cheng.

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
