## [Decision Letter · Decision Letter 0]

8 Nov 2024

PONE-D-24-35956The alteration of uterine microbiota participated in the activation of the decidual inflammatory response in early spontaneous abortionPLOS ONE

Dear Dr. Cheng, 

Thank you for submitting your manuscript to PLOS ONE. After careful consideration, we feel that it has merit but does not fully meet PLOS ONE’s publication criteria as it currently stands. Therefore, we invite you to submit a revised version of the manuscript that addresses the points raised during the review process.

After a thorough review, we would like to commend you on the quality of your work and its potential contribution to the study of early spontaneous abortion. Your study presents changes in uterine microbiota as a possible cause to early spontaneous abortion.

We believe that it is very close to being ready for publication. However, the peer-reviewers have identified a few minor revisions that would further strengthen the paper and improve its clarity. The peer-reviewed comments are attached herewith.

We are confident that addressing these points will help enhance the overall quality of the manuscript.

We appreciate your attention to these points and look forward to receiving your revised manuscript. Should you have any questions or need further clarification on any of the revisions, please do not hesitate to contact us.

We look forward to receiving your revised manuscript.

Kind regards,

Sameer Timilsina, MD

Academic Editor

PLOS ONE

Journal Requirements: When submitting your revision, we need you to address these additional requirements. 1. Please ensure that your manuscript meets PLOS ONE's style requirements, including those for file naming. The PLOS ONE style templates can be found at https://journals.plos.org/plosone/s/file?id=wjVg/PLOSOne_formatting_sample_main_body.pdf and https://journals.plos.org/plosone/s/file?id=ba62/PLOSOne_formatting_sample_title_authors_affiliations.pdf 2. We note that the grant information you provided in the ‘Funding Information’ and ‘Financial Disclosure’ sections do not match.  When you resubmit, please ensure that you provide the correct grant numbers for the awards you received for your study in the ‘Funding Information’ section. 3. Thank you for stating the following financial disclosure: "This research was funded by the National Natural Science Foundation of China (No.82101717)." Please state what role the funders took in the study.  If the funders had no role, please state: ""The funders had no role in study design, data collection and analysis, decision to publish, or preparation of the manuscript.""  If this statement is not correct you must amend it as needed. Please include this amended Role of Funder statement in your cover letter; we will change the online submission form on your behalf. 4. Please review your reference list to ensure that it is complete and correct. If you have cited papers that have been retracted, please include the rationale for doing so in the manuscript text, or remove these references and replace them with relevant current references. Any changes to the reference list should be mentioned in the rebuttal letter that accompanies your revised manuscript. If you need to cite a retracted article, indicate the article’s retracted status in the References list and also include a citation and full reference for the retraction notice.

Reviewers' comments:

Reviewer's Responses to Questions

**Comments to the Author**

1. Is the manuscript technically sound, and do the data support the conclusions?

Reviewer #1: Partly

Reviewer #2: Yes

2. Has the statistical analysis been performed appropriately and rigorously? 

Reviewer #1: Yes

Reviewer #2: Yes

3. Have the authors made all data underlying the findings in their manuscript fully available?

Reviewer #1: Yes

Reviewer #2: Yes

4. Is the manuscript presented in an intelligible fashion and written in standard English?

Reviewer #1: Yes

Reviewer #2: Yes

5. Review Comments to the Author

Reviewer #1: The study provides valuable insights into the relationship between uterine microbiota and inflammatory responses in early spontaneous abortion (ESA). However, some clarifications and adjustments will improve the manuscript’s clarity and robustness.

Figures and Tables:

Figures are informative, but some (e.g., Figures 3 and 4) are dense and could benefit from being split into sub-panels for clarity. Explain the Bayesian test with BH correction in the figure legend.

Methods:

Contamination Control: Provide more details on how contamination from the lower genital tract was controlled during 16S rRNA gene sequencing.

Cytokine Assays: Clarify the use of negative controls in cytokine detection.

Patient Selection: Specify the randomization method used for selecting ESA and AA patients.

Study Design:

Confounders: Clarify if lifestyle, diet, or other factors that affect microbiota were controlled.

Longitudinal Studies: Consider a longitudinal design to track microbiota and cytokine changes over time.

Mechanistic Studies: Future work could explore how microbial shifts drive inflammatory responses.

Statistics:

Multiple Comparisons: Clarify if corrections (e.g., Benjamini-Hochberg) were applied for multiple tests.

Missing Data: Explain how missing data were handled, especially in microbial sequencing and cytokine profiles.

Confounding Variables: Describe whether multivariate analyses were performed to control for confounders.

Results and data accessibility:

Patient-Level Data: Include more granular patient-level data in supplementary materials to strengthen the correlation between microbiota and inflammatory responses.

Outliers: Clarify how outliers in cytokine levels or microbiota data were handled.

References:

Include recent studies on reproductive tract microbiota to provide context, and reference the bioinformatics tools (e.g., USEARCH, Silva database) more thoroughly.

Title:

Consider simplifying the title.

Conclusions:

Emphasize the need for future mechanistic studies to explain how microbiota shifts influence inflammation in ESA.

Final Recommendation:

This is an important study. Clarifying the methods, refining figure presentation, and addressing statistical concerns will enhance the manuscript.

Overall Rating:

Significance:High

Clarity: Moderate (requires some clarification)

Reproducibility: Good (with minor adjustments)

Statistical Soundness:** Moderate (requires clarification)

Reviewer #2: This is a well-written manuscript with good quality research demonstrated. I would like to request the authors for a minor revision:

1. Immunological interaction in the decidual environment is crucial in terms of pregnancy success. The concept has been clearly elucidated. One factor is the HLA-G expression in the trophoblastic cells and the detection of soluble isoforms can guide predicting pregnancy outcomes (pre-eclampsia is one of the causes of spontaneous abortion). Below is a study worth citing:

Bhattarai A, Shah S, Dahal K, Neupane R, Thapa S, Neupane N, Barboza JJ, Shrestha A, Sah R, Apostolopoulos V. Biomarker role of maternal soluble human leukocyte antigen G in pre-eclampsia: A meta-analysis. Immun Inflamm Dis. 2024 Apr;12(4):e1254. doi: 10.1002/iid3.1254. PMID: 38639563; PMCID: PMC11027746.

Overall, I have no significant comments to the authors. When interrogating in the introduction, I would like to request authors to mention that their objective was to answer those questions, rather than simply making interrogations.

Congratulations on the good work!

6. PLOS authors have the option to publish the peer review history of their article (what does this mean? ). If published, this will include your full peer review and any attached files.

**Do you want your identity to be public for this peer review?** For information about this choice, including consent withdrawal, please see our Privacy Policy .

Reviewer #1: **Yes: ** Dr Manim Amatya

Reviewer #2: No

---

## [Author Response · Author response to Decision Letter 1]

12 Dec 2024

In response to the reviewers’ and editors’ comments, we have carefully revised the manuscript. The revisions are summarized below and detailed in the accompanying response letter:

1. Compliance with Journal Requirements

• Ensured the manuscript complies with PLOS ONE’s style requirements, including file naming conventions.

• Revised the ‘Funding Information’ and ‘Financial Disclosure’ sections for consistency and accuracy, explicitly stating the role of funders.

2. Figures and Tables

• Split original Figure 3 into two separate figures for better clarity and adjusted the font size of Figure 4 (now Figure 5) to enhance readability.

• Updated figure legends to explain the statistical methods used, including Bayesian tests with Benjamini-Hochberg correction.

3. Methodological Details

• Added a detailed description of contamination control during 16S rRNA sequencing and clarified the rationale for negative controls in cytokine assays.

• Addressed the issue of randomization, acknowledging the observational nature of our study.

4. Addressing Confounding Variables

• Clarified the use of multivariate logistic regression to control for confounding factors and added results demonstrating adjusted associations in S1 Table .

5. Patient-Level Data and Missing Data

• Acknowledged limitations in subgroup sample size and addressed how missing data were handled. Future studies will expand sample size to strengthen findings.

6. Updated References

• Incorporated recent studies on reproductive tract microbiota and relevant bioinformatics tools USEARCH to provide additional context.

7. Title and Conclusion

• Retained the original title for clarity and relevance, while emphasizing the need for future mechanistic studies in the conclusions.

---

## [Decision Letter · Decision Letter 1]

2 Jan 2025

The alteration of uterine microbiota participated in the activation of the decidual inflammatory response in early spontaneous abortion

PONE-D-24-35956R1

Dear Dr. Cheng,

We’re pleased to inform you that your manuscript has been judged scientifically suitable for publication and will be formally accepted for publication once it meets all outstanding technical requirements.

Kind regards,

Sameer Timilsina, MD

Academic Editor

PLOS ONE

Additional Editor Comments (optional):

Thank you for submitting your manuscript to PLOS One. After peer-review we would like to inform you that your manuscript has been accepted for publication. Congratulations!!

Reviewers' comments:

Reviewer's Responses to Questions

**Comments to the Author**

1. If the authors have adequately addressed your comments raised in a previous round of review and you feel that this manuscript is now acceptable for publication, you may indicate that here to bypass the “Comments to the Author” section, enter your conflict of interest statement in the “Confidential to Editor” section, and submit your "Accept" recommendation.

Reviewer #1: All comments have been addressed

Reviewer #2: All comments have been addressed

2. Is the manuscript technically sound, and do the data support the conclusions?

Reviewer #1: Yes

Reviewer #2: Yes

3. Has the statistical analysis been performed appropriately and rigorously? 

Reviewer #1: Yes

Reviewer #2: I Don't Know

4. Have the authors made all data underlying the findings in their manuscript fully available?

Reviewer #1: Yes

Reviewer #2: Yes

5. Is the manuscript presented in an intelligible fashion and written in standard English?

Reviewer #1: Yes

Reviewer #2: Yes

6. Review Comments to the Author

Reviewer #1: (No Response)

Reviewer #2: (No Response)

7. PLOS authors have the option to publish the peer review history of their article (what does this mean? ). If published, this will include your full peer review and any attached files.

**Do you want your identity to be public for this peer review?** For information about this choice, including consent withdrawal, please see our Privacy Policy .

Reviewer #1: **Yes: ** Dr Manim Amatya, MD Pathology, Grande International Hospital, Dhapasi, Kathmandu, Nepal

Reviewer #2: No

---

## [Editor Report · Acceptance letter]

PONE-D-24-35956R1

PLOS ONE

Dear Dr. Cheng,

I'm pleased to inform you that your manuscript has been deemed suitable for publication in PLOS ONE. Congratulations! Your manuscript is now being handed over to our production team.

Kind regards,

on behalf of

Dr. Sameer Timilsina

Academic Editor

PLOS ONE